# Photocatalytic Reduction of Hexavalent Chromium Using Cu$_{3.21}$Bi$_{4.79}$S$_9$/g-C$_3$N$_4$ Nanocomposite

Timothy O. Ajiboye [1,2], Opeyemi A. Oyewo [3], Riadh Marzouki [4,5] and Damian C. Onwudiwe [1,2,*]

1    Material Science Innovation and Modelling (MaSIM) Research Focus Area, Faculty of Natural and Agricultural Sciences, North-West University, Mafikeng Campus, Private Bag X2046, Mmabatho 2735, South Africa

2    Department of Chemistry, School of Physical and Chemical Sciences, Faculty of Natural and Agricultural Sciences, North-West University, Mafikeng Campus, Private Bag X2046, Mmabatho 2735, South Africa

3    Department of Science and Technology Education, University of Johannesburg, Johannesburg 2092, South Africa

4    Chemistry Department, College of Science, King Khalid University, Abha 61413, Saudi Arabia

5    Chemistry Department, Faculty of Sciences of Sfax, University of Sfax, Sfax 3029, Tunisia

*    Correspondence: damian.onwudiwe@nwu.ac.za; Tel.: +27-18-389-2545; Fax: +27-18-389-2420

**Abstract:** The photocatalytic reduction of hexavalent chromium, Cr(VI), to the trivalent species, Cr(III), has continued to inspire the synthesis of novel photocatalysts that are capable of achieving the task of converting Cr(VI) to the less toxic and more useful species. In this study, a novel functionalized graphitic carbon nitride (Cu$_{3.21}$Bi$_{4.79}$S$_9$/gC$_3$N$_4$) was synthesized and characterized by using X-ray diffraction (XRD), thermogravimetry analysis (TGA), energy-dispersive X-ray spectroscopy (EDS), Fourier transform infrared spectroscopy (FTIR), transmission electron microscope (TEM), and scanning electron microscope (SEM). The composite was used for the photocatalytic reduction of hexavalent chromium, Cr(VI), under visible light irradiation. A 92.77% efficiency of the reduction was achieved at pH 2, using about 10 mg of the photocatalyst and 10 mg/L of the Cr(VI) solution. A pseudo-first-order kinetic study indicated 0.0076 min$^{-1}$, 0.0286 min$^{-1}$, and 0.0393 min$^{-1}$ rate constants for the nanoparticles, pristine gC$_3$N$_4$, and the nanocomposite, respectively. This indicated an enhancement in the rate of reduction by the functionalized gC$_3$N$_4$ by 1.37- and 5.17-fold compared to the pristine gC$_3$N$_4$ and Cu$_{3.21}$Bi$_{4.79}$S$_9$, respectively. A study of how the presence of other contaminants including dye (bisphenol A) and heavy-metal ions (Ag(I) and Pb(II)) in the system affects the photocatalytic process showed a reduction in the rate from 0.0393 min$^{-1}$ to 0.0019 min$^{-1}$ and 0.0039 min$^{-1}$, respectively. Finally, the radical scavenging experiments showed that the main active species for the photocatalytic reduction of Cr(VI) are electrons (e$^-$), hydroxyl radicals (·OH$^-$), and superoxide (·O$_2$$^-$). This study shows the potential of functionalized gC$_3$N$_4$ as sustainable materials in the removal of hexavalent Cr from an aqueous solution.

**Keywords:** graphitic carbon nitride; photoreduction; dithiocarbamate complexes; heavy-metal ions; nanocomposites



## 1. Introduction

Chromium is used in catalysis, leather tanning, electroplating, metal finishing, and also in the manufacturing of electronic devices, pigments, and magnetic tapes [1]. The vast applications of chromium have raised concerns about their fate and impacts on the environment [2]. It has the ability to accumulate in the food chain and enter into the systems of humans and animals, where it causes damage to vital organs [3]. Chromium exists in different oxidation states, and among these are the poisonous hexavalent chromium and the trivalent chromium, which are useful as micronutrients for plants [4]. The hexavalent chromium is stable and does not easily precipitate out of environmental samples, while

trivalent chromium easily precipitates out under alkaline conditions [5]. Hence, the reduction of toxic hexavalent chromium to useful trivalent chromium is a tenable approach for the removal of hexavalent chromium [6]. The reduction reaction could be inspired by different processes, and photocatalysis has emerged as a sustainable approach because it relies on the utilization of naturally available light absorbed by the photocatalyst.

The efficiency of the photocatalytic process depends on the effectiveness of the photocatalyst used [7–9]. Consequently, several photocatalysts have been investigated for the process. For instance, metallic [10], binary or ternary metal chalcogenides [11], doped materials (such as $Mn^{2+}$-doped CuS [12] and Ag-doped CoO [13]) nanoparticles, and heterojunction systems, including $BaFe_2O_4/SnO_2$ [14], $In_2S_3$-$ZnIn_2S_4$ [15], $TiO_2/MoS_2$ [16], and $WO_3/In_2S_3$ [17], have been used for the photocatalytic Cr(VI) reduction. The incorporation of nanoparticles into the covalent organic framework has been investigated for various applications including photocatalysis due to their excellent electron conductivity [18,19]. Polymeric materials such as graphitic carbon nitride have also been explored. Graphitic carbon nitride is a non-metallic photocatalyst which has attracted attention due to its inertness, stability, and ability to absorb light in the visible region of the solar spectrum [7,20]. The performance of graphitic carbon nitride has been functionalized with different materials to form composite photocatalysts. Examples of functionalized graphitic carbon nitride reported for the photocatalytic reduction of hexavalent chromium are $ZnIn_2S_4/g$-$C_3N_4$ [21] and g-$C_3N_4/Bi_3NbO_7$ [22].

Dithiocarbamate complexes are precursors for synthesizing metal sulphides because parameters such as the time of reaction, temperature of reaction, and solvent used could be tuned to obtain different morphological phases of the nanoparticles [23]. The compounds have been used for the synthesis of binary [24] and ternary sulphides. Copper bismuth sulphide is one of the numerous ternary sulphides that have been synthesized via the thermal decomposition of dithiocarbamate complexes [25,26].

Several non-fractional phases of copper bismuth sulphide ($Cu_3BiS_3$, $Cu_{24}Bi_{26}S_{51}$, $Cu_4Bi_7S_{12}$, $Cu_6Bi_4S_9$, $CuBiS_2$, and $CuBi_5S_8$ have been reported [8,27–29]. The non-common fractional stoichiometric phase ($Cu_{3.21}Bi_{4.79}S_9$) also exists, and Barma et al. has reported its synthesis via mechanical alloying [30]. The synthesis of the non-fractional stoichiometric phases has been reported by the solvothermal route [8,26]. However, to the authors' best of knowledge, no report exists on the synthesis of the fractional phase via solvothermal route. Hence, the present study reports the synthesis of the fractional phase of copper bismuth sulphide ($Cu_{3.21}Bi_{4.79}S_9$) through a solvothermal method. The *N*-methyl-*N*-phenyl dithiocarbamate complexes of copper and bismuth were used as the precursor compounds. The synthesis of these dithiocarbamate complexes were carried out via already reported procedure with slight modifications [31]. The as-synthesized ternary nanoparticles were used to functionalize graphitic carbon nitride to produce a novel composite photocatalyst ($Cu_{3.21}Bi_{4.79}S_9/gC_3N_4$). The nanocomposite was utilized for the photocatalytic reduction of hexavalent chromium under visible light.

## 2. Results and Discussion

### 2.1. X-ray Diffraction (XRD) Studies

The XRD pattern (Figure 1c) showed the formation of graphitic carbon nitride functionalized with $Cu_{3.21}Bi_{4.79}S_9$ without the appearance of peaks corresponding to metallic bismuth, copper sulphides, or bismuth sulphides. The two peaks at 2 theta values of 13.2° and 27.5° emanated from the graphitic carbon nitride and are indexed to the (100) and (002) planes, respectively (Figure 1a). The (100) peak is ascribed to the in-planar structural packing of tris-s-triazine, while the (002) peak is because of the interlayer stacking of the aromatic conjugated systems [7,32–34]. Apart from the peaks indexed for the graphitic carbon nitride, other major peaks in the pattern indexed to (202), (−204), (−112), (−403), (−312), (311), (−514), and (−316) planes corresponded to the base-centred monoclinic structure (JCPDS card No_01-073-1202, space group C2/m (12) (Figure 1b). The existence

of these peaks confirmed the formation of graphitic carbon nitride functionalized with $Cu_{3.21}Bi_{4.79}S_9$.

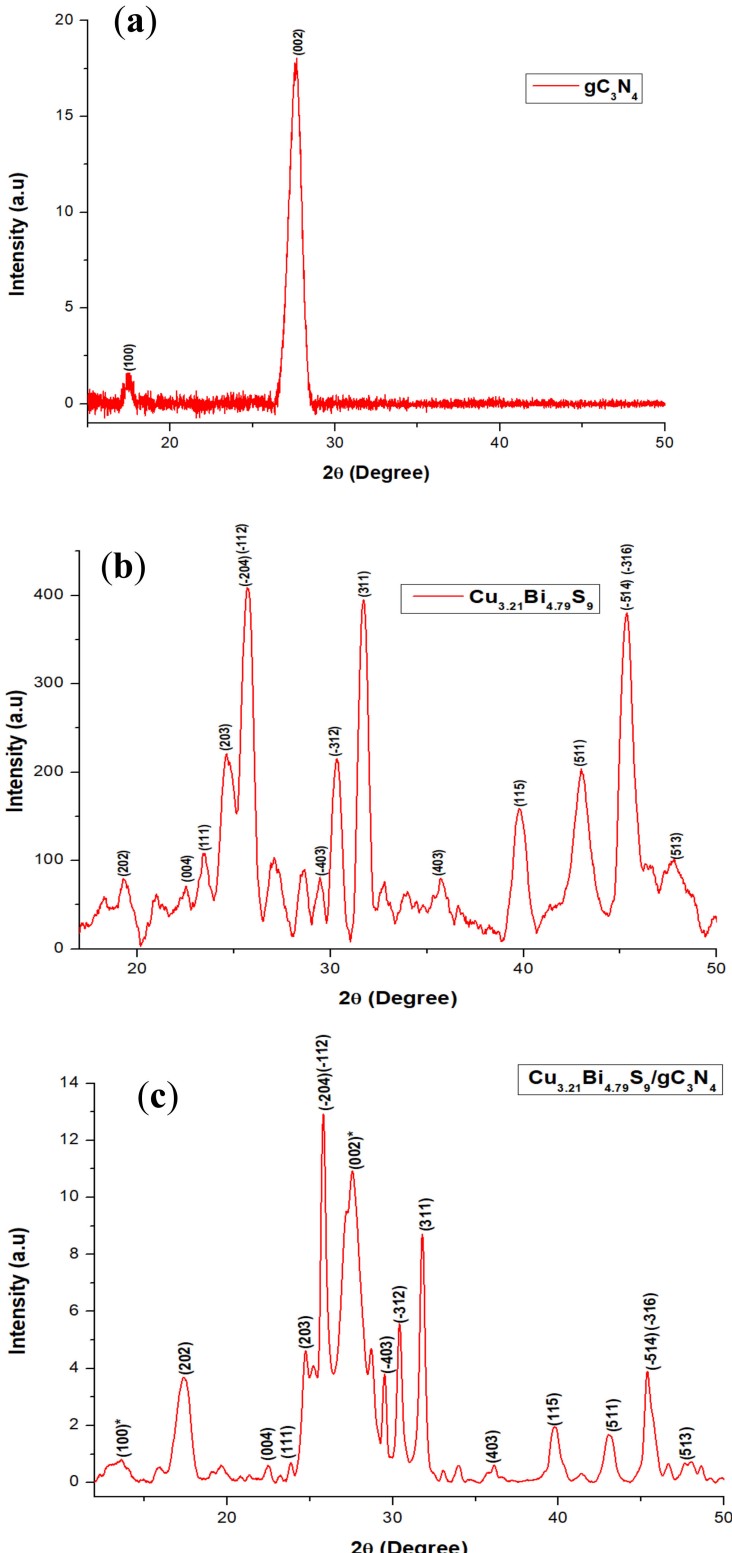

**Figure 1.** XRD pattern of (**a**) graphitic carbon nitride ($gC_3N_4$), (**b**) copper bismuth sulphide ($Cu_{3.21}Bi_{4.79}S_9$), and (**c**) $Cu_{3.21}Bi_{4.79}S_9/gC_3N_4$. The peaks of graphitic carbon nitride are asterisked (*).

## 2.2. FTIR Studies

Figure A1 is the FTIR spectrum of pristine copper bismuth sulphide. The peaks due the NH$_2$ bending mode (873 cm$^{-1}$), NH$_2$ scissor mode (1429 cm$^{-1}$), C-N stretch (1009 cm$^{-1}$), C-H bending mode (1569 cm$^{-1}$), and C-H stretching mode (2920–2669 cm$^{-1}$) were all from the oleylamine (used as the surfactant) [35]. These peaks were pronounced before the incorporation of copper bismuth sulphide. The FTIR of pristine graphitic carbon nitride and Cu$_{3.21}$Bi$_{4.79}$S$_9$/gC$_3$N$_4$- are presented in Figure 2. Both spectra showed a characteristic peak at 3128 cm$^{-1}$ due to the presence of N-H bonds. Additionally, the peaks between 1225 and 1629 cm$^{-1}$ appeared in both spectra due to the presence of –CN bonds for the heterocyclic ring. The last peak that is commonly observed for them appeared at 799 cm$^{-1}$, and this is attributed to the heptazine rings in the structure of graphitic carbon nitride and the nanocomposite [34,36]. In addition to these peaks, the peaks at 603 and 1104 cm$^{-1}$ are observable in the spectra of the nanocomposite, but they are absent in the pristine graphitic carbon nitride. The peak at 603 cm$^{-1}$ can be attributed to the Cu-S bond [37], while the peak at 1104 cm$^{-1}$ can be attributed to the Bi-S stretching vibration [38]. This further showed that the ternary copper bismuth sulphide was incorporated into the graphitic carbon nitride. It was observed that the peaks for the oleylamine were absent after the formation of the nanocomposite.

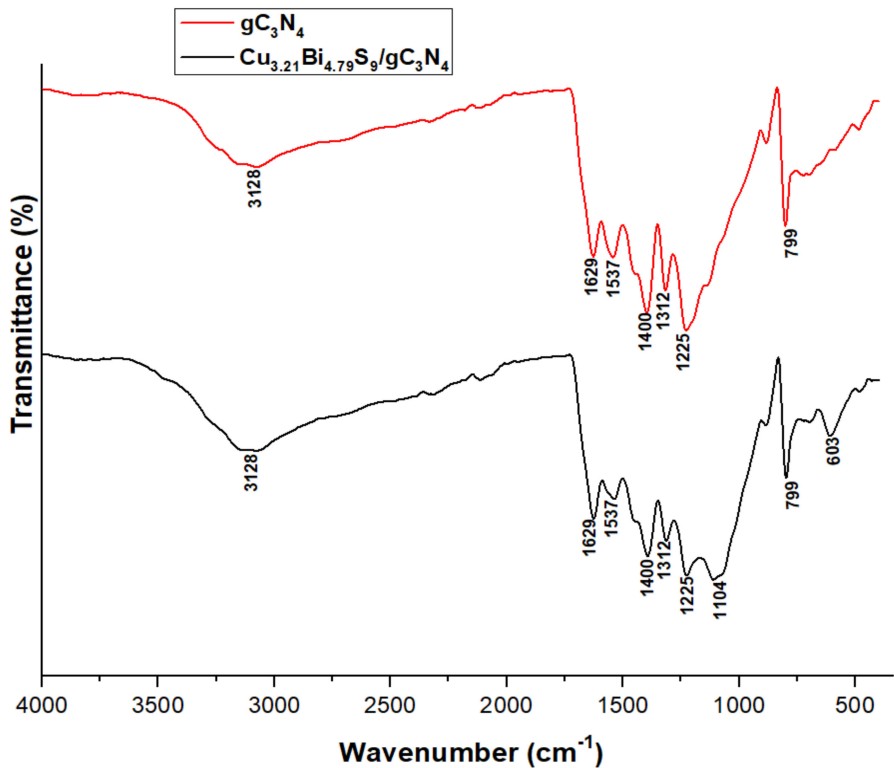

**Figure 2.** FTIR spectra of gC$_3$N$_4$ and Cu$_{3.21}$Bi$_{4.79}$S$_9$/gC$_3$N$_4$ nanocomposite.

## 2.3. Morphological Properties of Cu$_{3.21}$Bi$_{4.79}$S$_9$ and Cu$_{3.21}$Bi$_{4.79}$S$_9$/gC$_3$N$_4$

The SEM images of Cu$_{3.21}$Bi$_{4.79}$S$_9$ and Cu$_{3.21}$Bi$_{4.79}$S$_9$/gC$_3$N$_4$ presented in Figure 3a,b showed some agglomeration in their surface morphology. The level of agglomeration was higher in the pristine Cu$_{3.21}$Bi$_{4.79}$S$_9$ compared to Cu$_{3.21}$Bi$_{4.79}$S$_9$/gC$_3$N$_4$. Similarly, there was more uniformity in the particles obtained from the pristine copper bismuth sulphide compared to those from the nanocomposite. Hence, there was noticeable change in the surface morphology after the incorporation of graphitic carbon nitride to the pristine Cu$_{3.21}$Bi$_{4.79}$S$_9$. The EDX analysis (Figure 3c,d) confirmed the presence of copper, bismuth, and sulphur in the Cu$_{3.21}$Bi$_{4.79}$S$_9$.

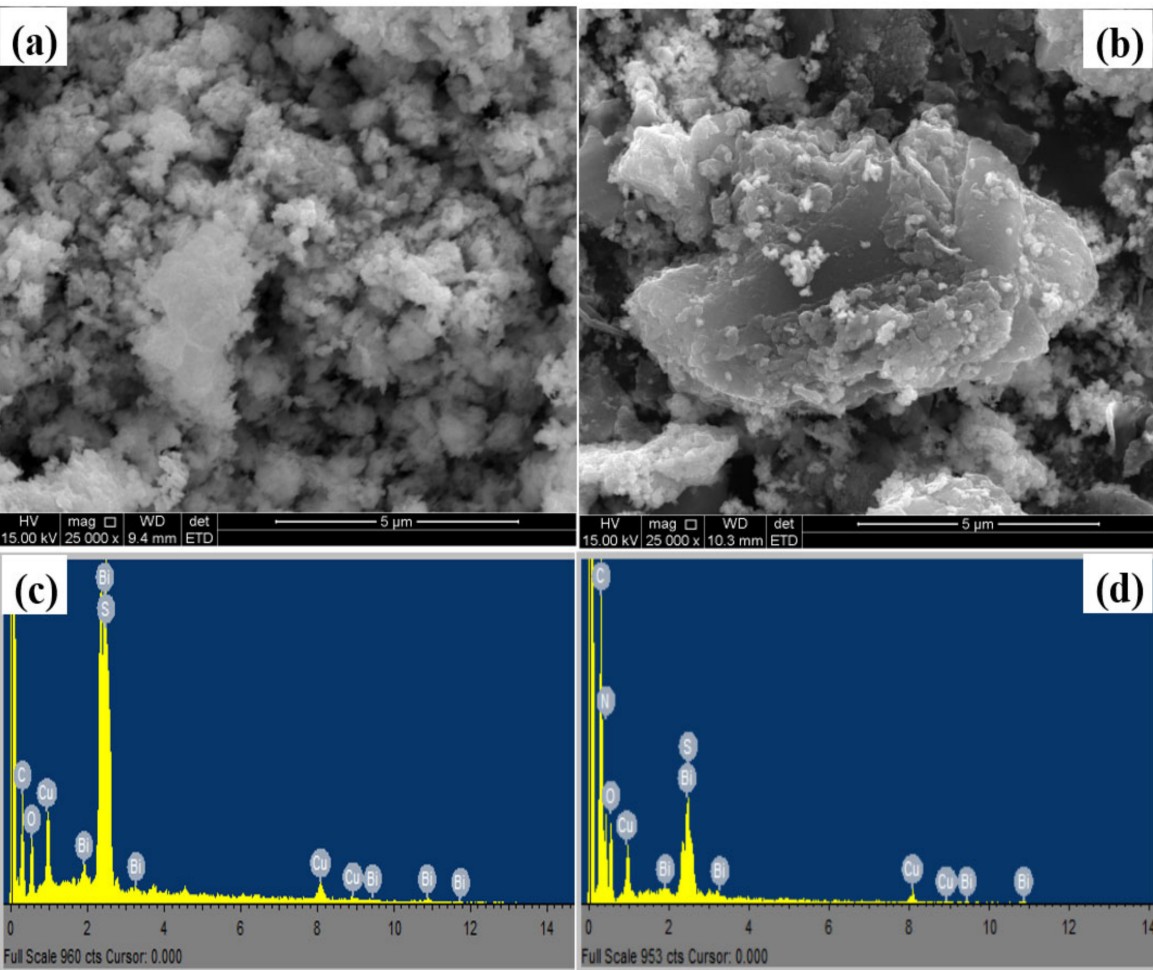

**Figure 3.** SEM images of (**a**) $Cu_{3.21}Bi_{4.79}S_9$, (**b**) $Cu_{3.21}Bi_{4.79}S_9/gC_3N_4$, and their respective (**c**,**d**) EDX spectra.

The TEM images of the $Cu_{3.21}Bi_{4.79}S_9$ nanoparticles and the $Cu_{3.21}Bi_{4.79}S_9/gC_3N_4$ nanocomposite are shown in Figure 4. The internal morphology of $Cu_{3.21}Bi_{4.79}S_9$ confirmed that they are rods (Figure 4a), and the HRTEM image (Figure 4b) revealed distinct lattice fringes that indicated they are crystalline materials, and the SAED image of Figure 4c shows the ring-like electron diffraction patterns consistent with crystalline nanorods. The microstructure of the nanocomposite showed rod-like structures embedded within a sheet-like aggregate (Figure 4d). The sheet-like structure was due to the presence of graphitic carbon nitride in the nanocomposites [39]. From the TEM micrograph, the rods had a length of 190.0 nm and breadth of 14.4 nm, as presented in the particle size distribution histograms (Figure 4e,f). The HRTEM image revealed that the d-spacing of the nanocomposite was 0.5206 nm along the (311) plane of $Cu_{3.21}Bi_{4.79}S_9$ in the nanocomposite (Figure 4g). In the pristine $Cu_{3.21}Bi_{4.79}S_9$, the d-spacing of 0.3822 nm was along the (311) plane (Figure A2). These d-spacings were obtained from Bragg's equation, and there was a noticeable increase in the d-spacing of the nanocomposite compared with that of the pristine $Cu_{3.21}Bi_{4.79}S_9$. The elemental mapping (Figures A3 and A4) further revealed that the copper bismuth sulphide was well-dispersed across the matrices of the graphitic carbon nitride in the nanocomposite.

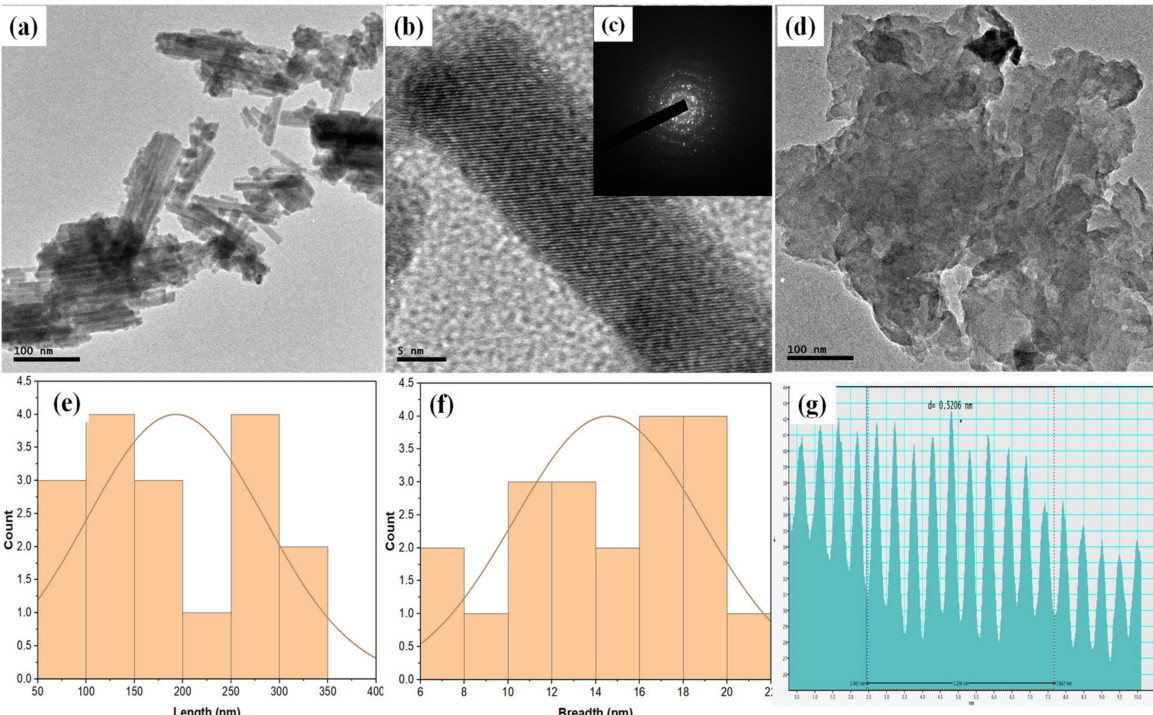

**Figure 4.** (**a**)TEM, (**b**) HRTEM, and (**c**) SAED images of $Cu_{3.21}Bi_{4.79}S_9$; (**d**) TEM image of $Cu_{3.21}Bi_{4.79}S_9/gC_3N_4$; particle size distribution histogram of $Cu_{3.21}Bi_{4.79}S_9$, showing (**e**) length, (**f**) breadth, and (**g**) is the d-spacing of the encapsulated nanoparticles.

### 2.4. Optical Properties

Figure 5 presents the overlaid absorption spectra of $gC_3N_4$, $Cu_{3.21}Bi_{4.79}S_9$, and $Cu_{3.21}Bi_{4.79}S_9/gC_3N_4$. The spectrum of $gC_3N_4$ showed its characteristic absorption maximum at 337 nm due to unsaturated n-$\pi$* transitions arising from the nitrogen atoms of the terminal –NH$_2$, polyheptazine, and polytriazine structural units. Another characteristic absorption maximum appeared at 394 nm, and it is attributed to the $\pi - \pi$* transition of conjugated heterocyclic aromatic components of graphitic carbon nitride (Figure 5a). This transition also accounts for the pale-yellow colouration of graphitic carbon nitride [33,40]. The maximum absorption for $Cu_{3.21}Bi_{4.79}S_9$ was found at 218 nm, unlike in the non-fractional phase ($Cu_3BiS_3$), whose absorption maximum was reported to be at 468 nm [27]. The absorption maximum of $Cu_{3.21}Bi_{4.79}S_9$ shifted to 225 nm after compositing with graphitic carbon nitride. The absorption peak of the graphitic carbon nitrides also appeared in the nanocomposites, but with a reduction in the intensities of these peaks. A Tauc plot was used to estimate the band gap energy of the semiconductor photocatalyst. The Tauc equation is given in Equation (1), where $\alpha$, $A$, $h$, $Eg$, $\nu$, $A$, and $n$ are the absorption coefficient, Planck's constant, optical band gap, proportionality constant, and Tauc exponent, respectively. The value of $n$ could be $\frac{1}{2}$, 3/2, 2, or 3 for direct (allowed) transitions, direct (forbidden) transitions, indirect (allowed) transition, or indirect (forbidden) transitions, respectively [41]. The indirect band gap energies of $gC_3N_4$, $Cu_{3.21}Bi_{4.79}S_9$, and $Cu_{3.21}Bi_{4.79}S_9/gC_3N_4$ were estimated by plotting $(ah\nu)^{1/2}$ against the energy of the photon ($h\nu$), and were obtained as 2.88, 0.55, and 1.25 eV, respectively (Figure 5b).

$$(\alpha h v)^{1/n} = A(hv - Eg) \tag{1}$$

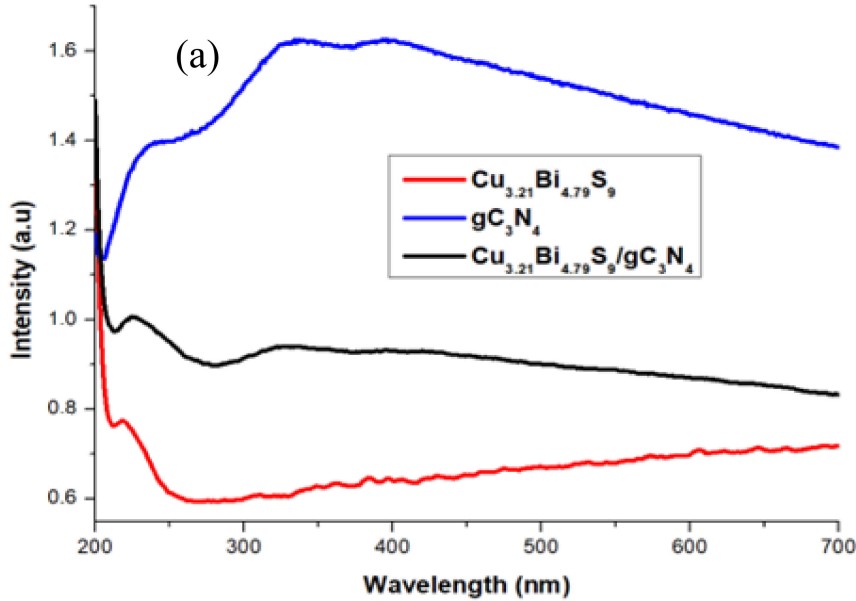

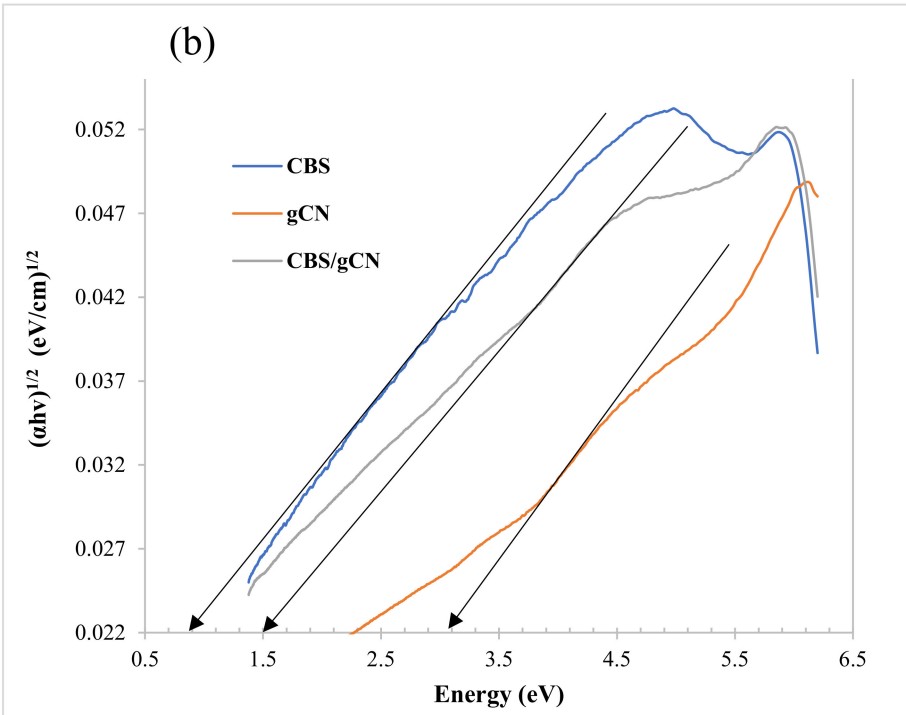

**Figure 5. (a)** Overlaid UV spectra of $Cu_{3.21}Bi_{4.79}S_9$, $gC_3N_4$, and $Cu_{3.21}Bi_{4.79}S_9/gC_3N_4$; **(b)** Tauc plot of $Cu_{3.21}Bi_{4.79}S_9$ (CBS), $gC_3N_4$ (gCN), and $Cu_{3.21}Bi_{4.79}S_9/gC_3N_4$ (CBS/gCN).

*2.5. Thermal Studies of $Cu_{3.21}Bi_{4.79}S_9/gC_3N_4$ Composite*

The TGA graph (Figure 6a) of the nanocomposite showed a slight weight loss at 72 °C due to the elimination of low-molecular-weight molecules (such as water and methanol) [42]. A pronounced weight loss occurred between 439 °C and 507 °C, which indicated the decomposition of graphitic carbon nitride. Complete volatilization of graphitic carbon nitride was observed at 677 °C. The decomposition profile was in agreement with the results obtained from the DSC graph (Figure 6b), which showed endothermic peaks at 71.97 °C and 676.65 °C, respectively. The high melting point of graphitic carbon nitride could be attributed to the presence of hydrogen bonding that exists between the strands of the polymeric melon units containing NH and $NH_2$ groups [43]. Even after the heating

temperature reached 1000 °C, the complete decomposition of the nanocomposite was not achieved because approximately 20% of the sample remained as residue and may be ascribed to the nanoparticles. This shows that the total volatilization of the nanocomposite is not feasible at 1000 °C due to the presence of the nanoparticles, which enhances the thermal stability of the nanocomposite.

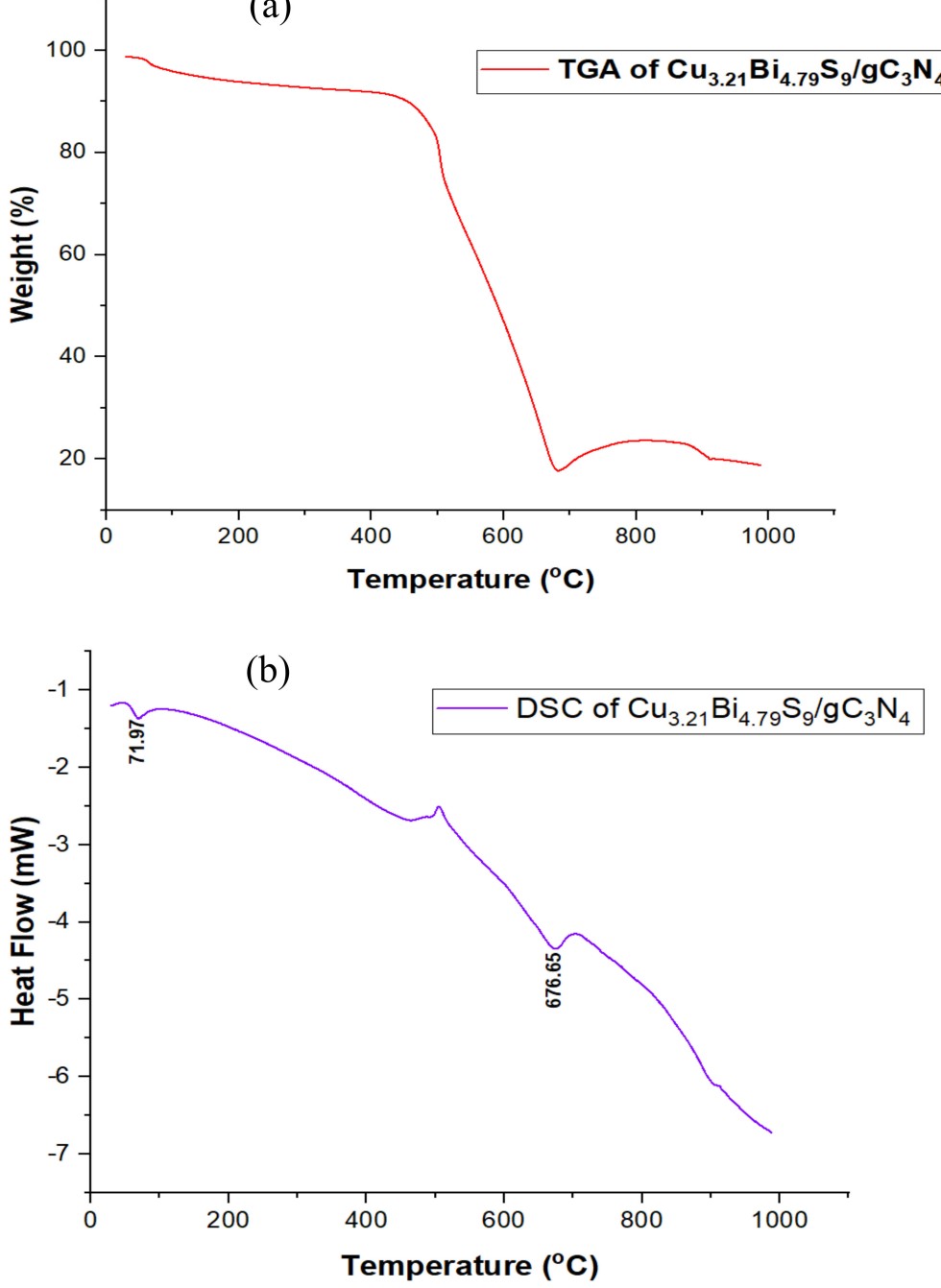

**Figure 6.** (**a**) TGA and (**b**) DSC of $Cu_{3.21}Bi_{4.79}S_9/gC_3N_4$ under nitrogen gas. TGA shows the thermal stability of the nanocomposite while DSC shows the thermal transitions.

### 3. Optimum Conditions for the Photocatalytic Investigations

The zeta potential of the nanocomposite was found to be highly negative in the alkaline pH ($-33.73$ mV) and under neutral pH ($-18.70$ mV), while it was positive under acidic conditions ($+12.95$ mV) (Figure 7). It has earlier been reported that hexavalent chromium exists as negatively charged $CrO_4^{2-}$ under acidic conditions. Since the surface

of the photocatalyst has positive zeta potential, there will be better hexavalent chromium adsorption under acidic conditions. This is because of the electrostatic force of attraction between the species and the surface of the photocatalyst. Consequently, the pH used for the photocatalytic reduction of chromium was pH 2. Based on previous reports, the concentration of Cr(VI) and the dosage of the photocatalyst chosen for the investigations were 10 mg/L and 10 mg, respectively [44–46].

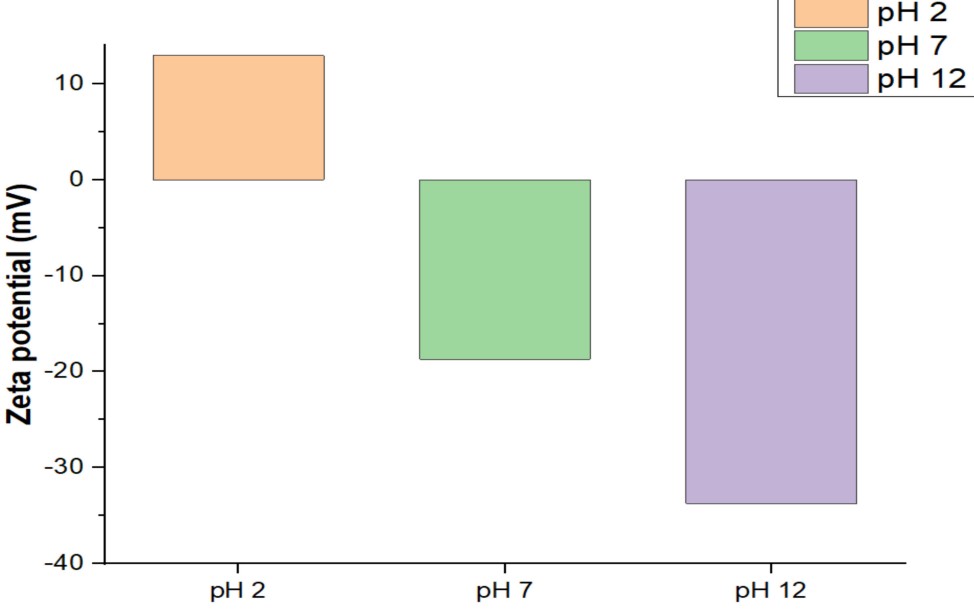

**Figure 7.** Zeta potential of $Cu_{3.21}Bi_{4.79}S_9/gC_3N_4$ at pH 2, 7, and 12.

### 3.1. Photocatalytic Investigations

The rate of the photocatalytic reduction of Cr(VI) was investigated by using the pristine $Cu_{3.21}Bi_{4.79}S_9$ nanorods, graphitic carbon nitride, and their nanocomposite as the photocatalyst under visible-light irradiation. As shown in (Figure 8a,b), the rate and percentage of the photocatalytic reduction of Cr(VI) were found to be 0.0076 min$^{-1}$ and 41.98%, respectively, when the pristine $Cu_{3.21}Bi_{4.79}S_9$ was used as the photocatalyst, while they were 0.0286 min$^{-1}$ and 78.57%, respectively, when the graphitic carbon nitride and the nanocomposite were used as the photocatalysts. The more than three-fold increase could be due to the polymeric nature of graphitic carbon nitride and its ability to adsorb Cr(VI) better than the $Cu_{3.21}Bi_{4.79}S_9$ photocatalyst [7,33,47]. Due to its better adsorption property, the photocatalytic performance was higher when graphitic carbon nitride was used as the photocatalyst [48,49]. There was a synergistic effect when the composite made from the two photocatalysts was used under visible light. The pseudo-first-order rate constant and percentage of the photocatalytic reduction increased to 0.0393 min$^{-1}$ and 92.77%, respectively. This shows an improvement in the photocatalytic rate of reduction of Cr(VI) by a factor of 1.37 and 5.17 compared to pristine graphitic carbon nitride and $Cu_{3.21}Bi_{4.79}S_9$, respectively. A similar synergistic photocatalytic effect was reported when graphitic carbon nitride was functionalized with other chalcogenides [50–52].

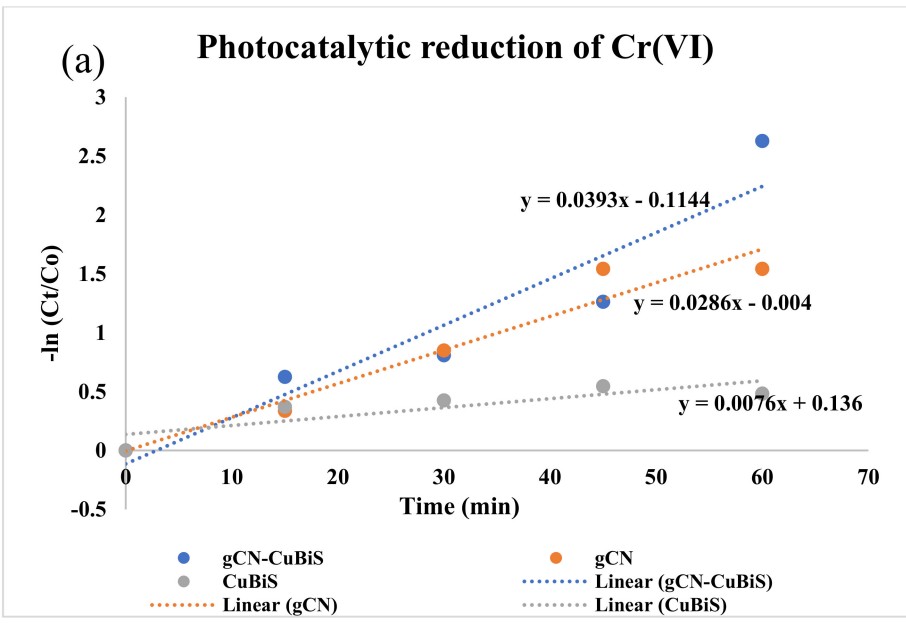

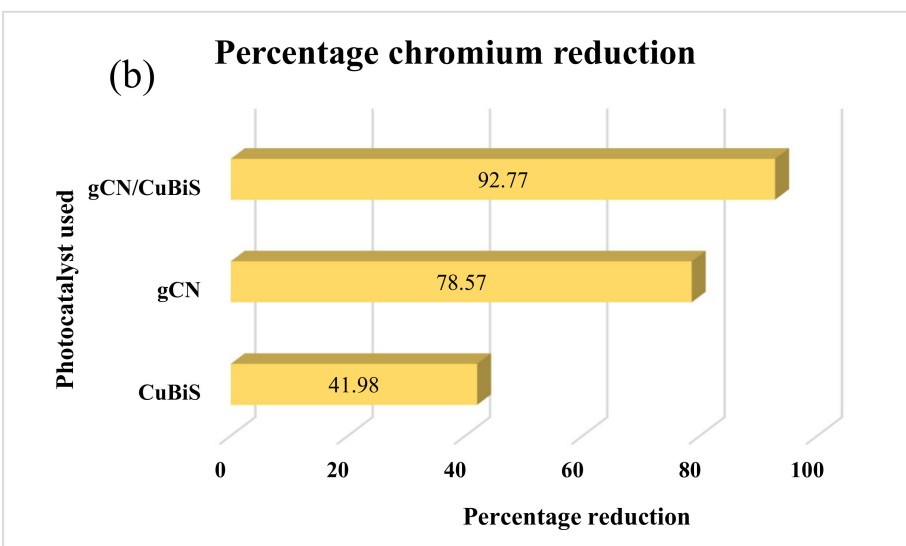

**Figure 8.** The (**a**) Pseudo-first-order rate and (**b**) percentage of photocatalytic reduction of Cr(VI) using pristine $Cu_{3.21}Bi_{4.79}S_9$, graphitic carbon nitride, and their composite.

### 3.2. Effect of Initial Temperature

The rate of the photocatalytic reduction of Cr(VI) was investigated at a very low temperature in ice (8 °C), at 25 and 50 °C, and the obtained results were compared. The rate was 0.0393 $min^{-1}$ under room-temperature conditions, while it dropped to 0.0077 and 0.0041 $min^{-1}$, respectively, in ice and at elevated temperature. The results show that the rate of the photocatalytic reduction was comparatively higher under room-temperature conditions than at higher and lower temperatures (Figure 9). Based on the kinetic postulates and the fact that the excitation process could make the change in Gibb's free energy (ΔG) lower than zero, the increase in temperature is expected to lead to an increasing rate of the photocatalytic process. However, higher temperature also leads to a drop in the photocurrent and higher rate of recombination of the photogenerated holes and electrons [53]. Furthermore, Gibb's free energy ΔG of the photocatalysts has been reported not to be affected by heat change [54]. The decrease in the rate of the photocatalytic reduction of Cr(VI) at high temperatures may be attributed to the reasons highlighted. Temperature increases can lead to the rate of the photocatalytic reduction of Cr(VI) if

there is enough thermodynamic driving force for electron transfer in the photocatalytic interface [53], as shown in Equation (2). Obviously, this cannot be attained at a temperature that is as low as 8 °C, which also accounts for the low photocatalytic reduction of Cr(VI).

$$K_{IT} = V_o e^{\frac{-Q}{KT}} \tag{2}$$

where $T$ is the thermal temperature, $K$ is the Boltzmann constant, $V_o$ is a pre-exponential factor, $Q$ is the apparent activation energy of the interfacial transfer, and $K_{IT}$ is the rate constant of interfacial transfer.

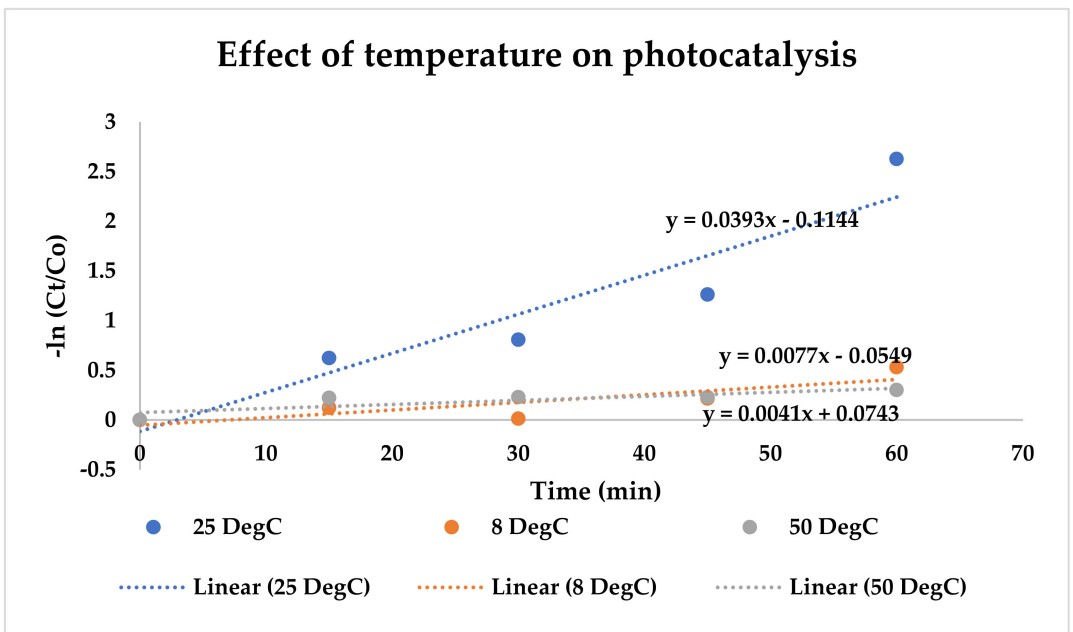

**Figure 9.** Photocatalytic reduction of Cr(VI) at varying temperatures.

### 3.3. Effect of the Presence of Bisphenol A

The effects of the presence of bisphenol A on the photocatalytic reduction of Cr(VI) were investigated. It was observed that the presence of bisphenol A reduced the rate of the photocatalytic reduction of Cr(VI) by more than 20-fold (from 0.0393 min$^{-1}$ to 0.0019 min$^{-1}$) (Figure 10). The bisphenol A in the photocatalytic system degraded with the pseudo-first-order rate constant of 0.0164 min$^{-1}$. The simultaneous degradation of bisphenol A with the photocatalytic reduction of Cr(VI) is in agreement with the report of the studies by Kim et al. [55]. It shows that Cr(VI) could be the electron scavenger of $Cu_{3.21}Bi_{4.79}S_9/gC_3N_4$-mediated photocatalytic bisphenol A degradation [56]. However, there was a retarded performance in the reduction of Cr(VI) in the presence of bisphenol A, contrary to the synergistic performance reported by Kim et al. [55]. The observation could be because bisphenol A is an aromatic compound and can resist photocatalytic degradation unlike the easily oxidizable organics such as oxalic acid, which enhances the rate of the photocatalytic process [57,58]. Due to the resistance to degradation, the possibility of electron capture by Cr(VI) is reduced since the positively charged holes are not scavenged by the added organics [59].

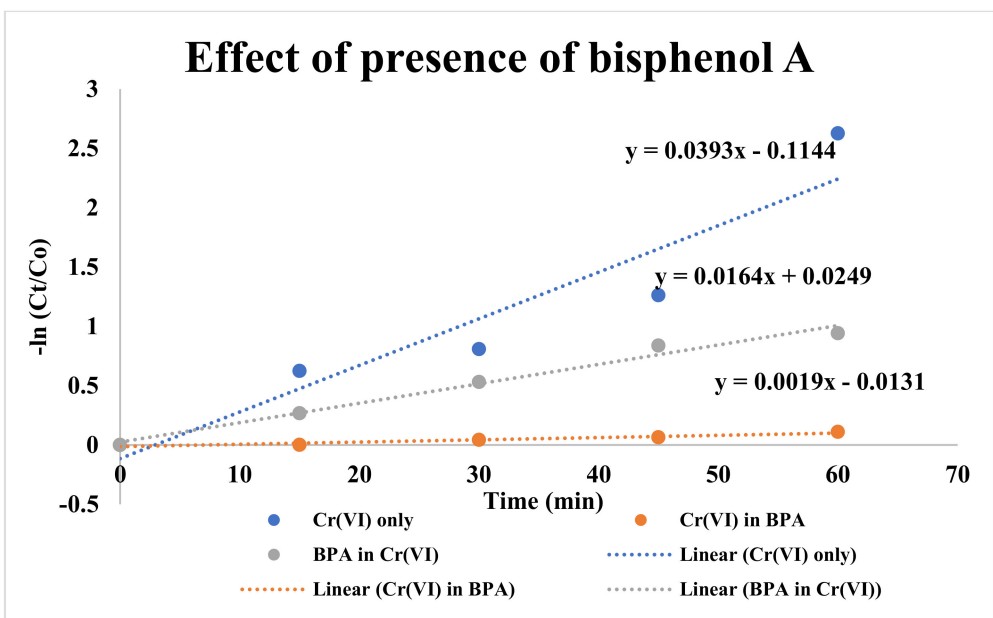

**Figure 10.** The photocatalytic reduction of Cr(VI) in the presence and absence of bisphenol A.

*3.4. Effects of the Presence of Other Heavy-Metal Ions*

The photocatalytic reduction of Pb(II) and Ag(I) was also investigated under similar photocatalytic conditions used for the reduction of Cr(VI) to Cr(III). This was for the purpose of studying how the presence of other ions affects the photoreduction process. The rate of the photocatalytic reduction of these heavy-metal ions was found to be low compared to that of Cr(VI). This could be because the surface of the photocatalyst is positively charged as well as Pb(II) and Ag(I), unlike Cr(VI), which exists as negatively charged $CrO_4^{2-}$ under the operating conditions. There was an electrostatic repulsion between the positively charged ions and the positively charged surface of the photocatalyst, while electrostatic attraction existed between the negatively charged $CrO_4^{2-}$ and the positively charged photocatalyst surface. Therefore, the adsorption of $CrO_4^{2-}$ to the surface of the photocatalyst was better than that of Pb(II) and Ag(I). So, the rate of the photocatalytic reduction was 0.0393, 0.0032, and 0.0004 $min^{-1}$ for Cr(VI), Ag(I), and Pb(II), respectively (Figure 11a). The rate of the photocatalytic reduction of Ag(I) was higher than that of Pb(II), and this could be ascribed to the reduction of monovalent silver, which requires one electron, while the reduction of divalent lead requires two electrons. Additionally, the standard reduction potential of $Ag^+/Ag$ is positive, while the standard reduction potential of $Pb^{2+}/Pb$ is negative (as shown in Equations (3)–(5)). The more positive the standard electrode potential, the more feasible the reaction [60]. The high rate of the photocatalytic reduction of Cr(VI) can also be attributed to the high positive potential of $Cr^{6+}/Cr^{3+}$.

$$Cr_2O_7^{2-} + 14H^+ + 6e^- \rightarrow 2Cr^{3+} + 7H_2O \qquad E° = +1.36 \text{ V} \qquad (3)$$

$$Ag^+ + e^- \rightarrow Ag_{(S)} \qquad E° = +0.80 \text{ V} \qquad (4)$$

$$Pb^{2+} + 2e^- \rightarrow Pb_{(S)} \qquad E° = -0.13 \text{ V} \qquad (5)$$

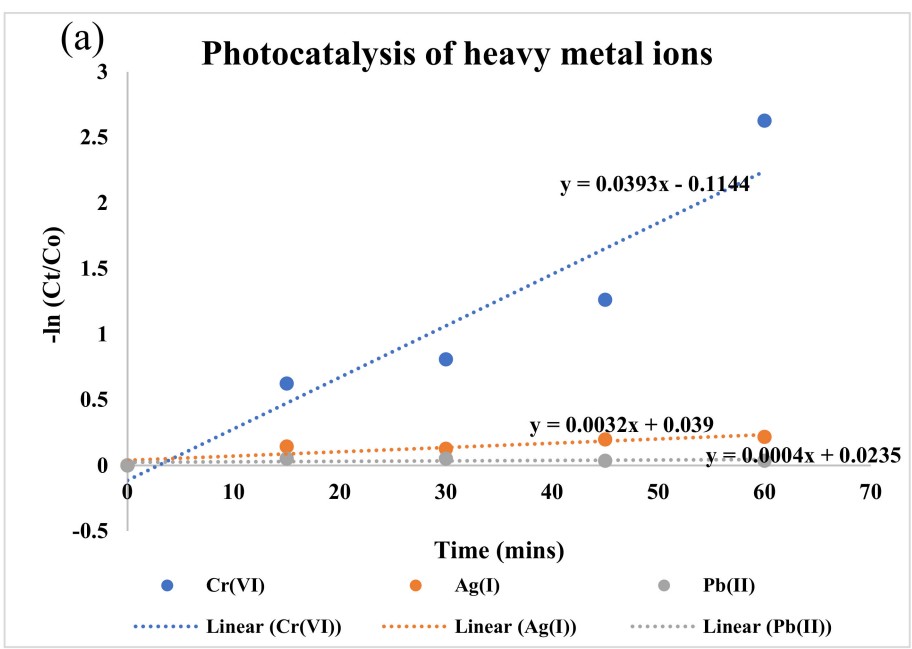

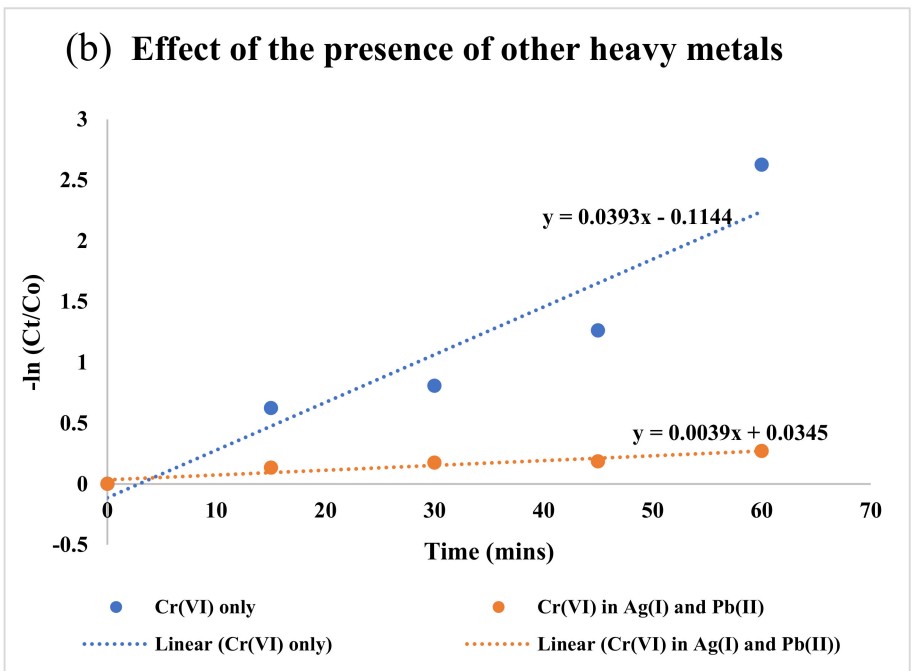

**Figure 11.** (**a**) Comparative studies of the photocatalytic reduction of Ag(I), Pb(II), and Cr(VI), (**b**) the photocatalytic reduction of Cr(VI) in the presence and absence of Ag(I) and Pb(II).

The rate of the photocatalytic reduction of Cr(VI) dropped from 0.0393 to 0.0039 $min^{-1}$ (half-life of 17.64 min to 177.73 min) when the mixture of Ag(I) and Pb(II) was introduced into the photocatalytic systems shown in Figure 11b. This clearly shows that the presence of other heavy-metal ions in the system containing Cr(VI) has antagonistic effects on the rate of the photocatalytic reduction of Cr(VI). It has been reported that cations such as Mg(II), Ca(II), K(I), and Na(I) have little effects on the photocatalytic reduction of Cr(VI) due to the high oxidation state and stability of Cr(VI) [61]. On the contrary, the present results show that the presence of heavy-metal ions such as Pb(II) and Ag(I) has a significant impact.

### 3.5. Radical Scavenging Experiment

The active species that participated in the photocatalytic reduction process were investigated using sodium nitrate, ascorbic acid (ASC), tert-butanol (TBA), and triethanol amine (TEA) as the scavenger for electrons ($e^-$), hydroxyl radicals ($\cdot OH^-$), superoxide ($\cdot O_2^-$ and holes ($h^+$), respectively. Without adding these scavengers, 92.77% of Cr(VI) was photocatalytically reduced over $Cu_{3.21}Bi_{4.79}S_9/gC_3N_4$ under visible light. With the addition of ASC and TBA into the photocatalytic system, the percentage of Cr(VI) that was photocatalytically reduced decreased from 92.77% to 8.2% and 11.04%, respectively (Figure 12). This shows that both hydroxyl radicals and holes were captured during the photocatalytic reduction process. This is in agreement with the report that the capturing of $\cdot O_2^-$ and $\cdot OH^-$ could improve the photocatalytic activities of $e^-$ in reducing Cr(VI) [62]. The introduction of sodium nitrate into the solution led to the drop in the percentage of photocatalytic reduction from 92.77% to 16.5%, which shows that $e^-$ plays a prominent role in the photocatalytic reduction of Cr(VI). Compared to other scavengers, the contribution of TEA is the lowest. However, there was also a significant decrease in the percentage of reduction because once the $h^+$ has been captured, the rate of the photocatalytic recombination is reduced, which allows the free $e^-$ to effect the reduction of Cr(VI) to Cr(III).

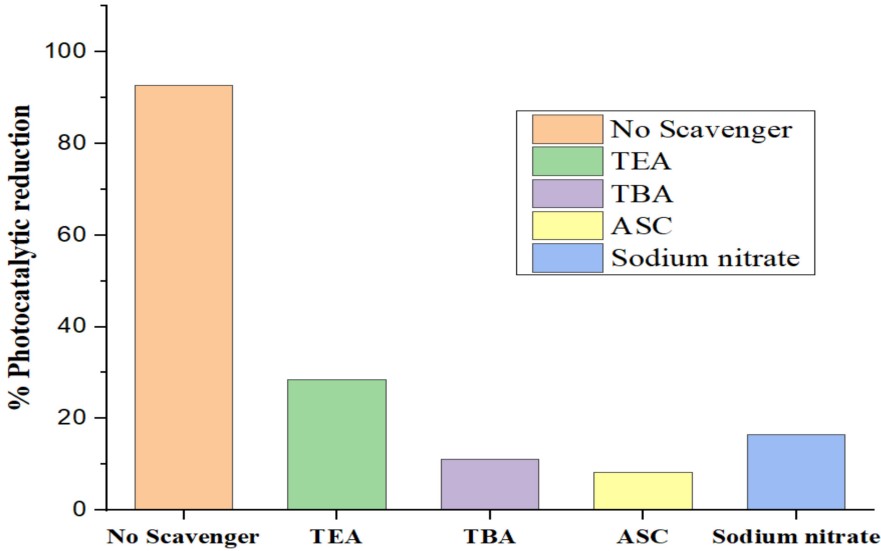

**Figure 12.** The results of the radical scavenging experiment.

## 4. Experimental

### 4.1. Materials

The melamine ($C_3H_6N_6$), oleylamine ($C_{18}H_{37}N$), hydrated bismuth(III) nitrate ($Bi(NO_3)_3 \cdot 5H_2O$)), copper(II) nitrate pentahydrate ($Cu(NO_3)_2 \cdot 5H_2O$), absolute ethanol ($CH_3CH_2OH$), and 2,4,6-trichlorophenol and chloroform ($CHCl_3$) used in this study were all procured from Merck chemicals company as analytical-grade reagents. Phillips X'pert diffractometer (fixed with single-wavelength Cu K$\alpha$ radiation ($\lambda$ = 1.546060 Å and operated at 40 kV/50 mA) was used for obtaining the X-ray diffraction (XRD) results. Perkin Elmer $\lambda$20 UV–vis spectrophotometer and PerkinElmer LS 45 fluorimeter were used to obtain the absorption and emission properties, respectively. Additionally, LYRA 3, TESCAN, and JEM—2100 JEOL equipment were used to obtain the scanning electron microscopy (SEM) and transmission electron microscopy (TEM) results, respectively. Thermogravimetry analysis was conducted using the SDTQ 600 V20.9 Build 20 Thermal analyser. Malvern Zetasizer Nanoseries was used for the zeta potential determination.

### 4.2. Synthesis of Graphitic Carbon Nitride (gC₃N₄)

About 8.0 g of melamine was transferred into an alumina crucible with a lid. The covered crucible was placed in the muffle furnace and heated at 550 °C for 4 h. The white

melamine powder was observed to change to a yellow solid. The obtained solid was cooled to room temperature and crushed with mortar and pestle.

### 4.3. Synthesis of N-Methyl-N-phenyl Dithiocarbamate Ligand and Complexes

The *N*-methyl-*N*-phenyl dithiocarbamate ligand was prepared by following an already reported procedure with slight modifications [31]. Briefly, *N*-methyl aniline (0.05 mol) was introduced into a round-bottom flask, and carbon disulphide (0.05 mol) was added. The solution was placed inside ice to maintain a very low temperature and stirred for about 15 min. This was followed by the addition of 15 mL of concentrated aqueous ammonia and was further stirred for 5 h. Light-yellow solid precipitate was obtained at the end of the reaction, and the obtained precipitate was filtered by suction and rinsed with 100 mL of ice-cold absolute ethanol.

The copper(II)-*N*-methyl-*N*-phenyl dithiocarbamate complex was prepared by reacting 20 mL of aqueous solution of the dithiocarbamate ligand (8.0 mmol) with 30 mL of ethanol solution of copper(II) nitrate hemipentahydrate (4.0 mmol) in a round-bottomed flask. The reaction was continued for 1 h at room temperature, forming deep-brown precipitates. The obtained product was filtered, rinsed with absolute ethanol, and recrystallized in a solution of chloroform. A similar procedure was used for the synthesis of the bismuth(III)- tris(*N*-methyl-*N*-phenyl dithiocarbamate) complex, but 12.0 mmol of the *N*-methyl-*N*-phenyl dithiocarbamate ligand was used against the 4.0 mmol of bismuth salt.

### 4.4. Synthesis of $Cu_{3.21}Bi_{4.79}S_9$

The $Cu_{3.21}Bi_{4.79}S_9$ nanoparticles were synthesized under nitrogen to prevent oxidation. 0.1069 g (0.25 mmol) of the copper (II) bis(*N*-methyl-*N*-phenyl dithiocarbamate) complex, 0.5663 g (0.75 mmol) of bismuth(III)- bis(*N*-methyl-*N*-phenyl dithiocarbamate) and 30 mL of oleylamine (capping agent) were introduced into three-necked round-bottom flask. The mixture was heated to 200 °C at a constant heating rate of 10 °C/min and stirred for 1 h. After the reaction, the flask was allowed to cool to 40 °C, and the addition of excess ethanol resulted in the precipitation of the nanoparticles. The products were separated from the solution by centrifugation at 4500 rpm for 5 min followed by decantation of the supernatant. The as-synthesized nanoparticles were washed four times with absolute ethanol and dried. The entire process is summarized in Scheme 1.

**Scheme 1.** Reaction process for the synthesis of (**a**) ligand, (**b**) complexes, and (**c**) copper bismuth sulphide ($Cu_{3.21}Bi_{4.79}S_9$) nanoparticles.

### 4.5. Synthesis of $Cu_{3.21}Bi_{4.79}S_9/gC_3N_4$

The graphitic carbon nitride and the as-prepared copper bismuth sulphide were homogenized in the ratio of 3:1 *w/w*, respectively. The homogenized sample was ground and calcined at 300 °C for 2 h in a muffle furnace to obtain $Cu_{3.21}Bi_{4.79}S_9/gC_3N_4$.

### 4.6. Photocatalytic Investigations

Reduction of aqueous solution of hexavalent chromium was evaluated by using a $Cu_{3.21}Bi_{4.79}S_9/gC_3N_4$ photocatalyst. In a typical synthesis procedure, 40 mL of 10 mg/L $K_2Cr_2O_7$ solution was introduced into a 100 mL beaker. Dilute hydrochloric acid was used to adjust the pH of the solution to 2. Afterwards, 10 mg of the photocatalyst was measured into the content in the beaker and stirred for 20 min in the dark to attain the adsorption–desorption equilibrium. This was immediately followed by the irradiation with 48 W visible LED light. During the reaction, aliquots were taken at 15 min regular intervals. The concentration of each aliquot was determined by using UV–visible spectrophotometer. The percentage of Cr(VI) that was photocatalytically reduced was estimated by using Equation (6), while the rate of the photocatalytic reduction was obtained via pseudo-first-order kinetics given in Equation (7). The half-life was obtained from the rate constant by using Equation (8).

$$Percentage\ photocatalytic\ reduction = \frac{C_o - C_t}{C_o}\ 100\%. \tag{6}$$

$$-Kt = \ln(\frac{C_t}{C_o}) \tag{7}$$

$$Half - life = {}^{\ln 2}/k = {}^{0.693}/k \tag{8}$$

where $C_o$ and $C_t$ represent the concentration of aqueous Cr(VI) at the initial time and at a specific time, $t$ is the reaction time, and $k$ is the pseudo-first-order reaction rate constant.

## 5. Conclusions

In general, the graphitic carbon nitride functionalized with the rod-like $Cu_{3.21}Bi_{4.79}S_9$ photocatalyst displayed a good photocatalytic reduction efficiency for Cr(VI) under visible-light irradiation. The results reveal that over 90% of Cr(VI) was photocatalytically reduced within 1 h at a pH of 2, photocatalyst dosage of 10 mg, and Cr(VI) concentration of 10 mg/L. The presence of bisphenol A, Ag(I), and Pb(II) in the photocatalytic system had inhibitory effects on the photocatalytic reduction of Cr(VI). The radical scavenging experiments revealed that electrons ($e^-$), hydroxyl radicals ($\cdot OH^-$), and superoxide ($\cdot O_2^-$) played significant roles in the photocatalytic reduction of Cr(VI) in the presence of the as-prepared photocatalyst under visible light. A pseudo-first-order kinetic study indicated 0.0076 $min^{-1}$, 0.0286 $min^{-1}$, and 0.0393 $min^{-1}$ rate constants for the nanoparticles, pristine $gC_3N_4$, and the nanocomposite, respectively. This indicated an enhancement in the rate of reduction by the functionalized graphitic carbon nitride by 1.37- and 5.17-fold compared to the pristine $gC_3N_4$ and $Cu_{3.21}Bi_{4.79}S_9$, respectively.

**Author Contributions:** Conceptualization, D.C.O.; methodology, T.O.A.; validation, T.O.A.; formal analysis, T.O.A., D.C.O., R.M. and O.A.O.; investigation, T.O.A. and R.M.; writing—original draft preparation, T.O.A.; writing—review and editing, D.C.O.; visualization, T.O.A. and D.C.O.; supervision, D.C.O. and O.A.O. All authors have read and agreed to the published version of the manuscript.

**Funding:** Authors extend their appreciation to the Deanship of Scientific Research at King Khalid University for funding this work through research group project under grant number R.G.P-2/175/43 and also gratefully acknowledge the financial support from the North-West University South Africa (grants ref: 1K02799).

**Acknowledgments:** The Authors extend their appreciation to the Deanship of Scientific Research at King Khalid University for funding this work through research group under grant number R.G.P-2/175/43, and the North-West University, South Africa.

**Conflicts of Interest:** The authors declare no conflict of interest.

## Appendix A

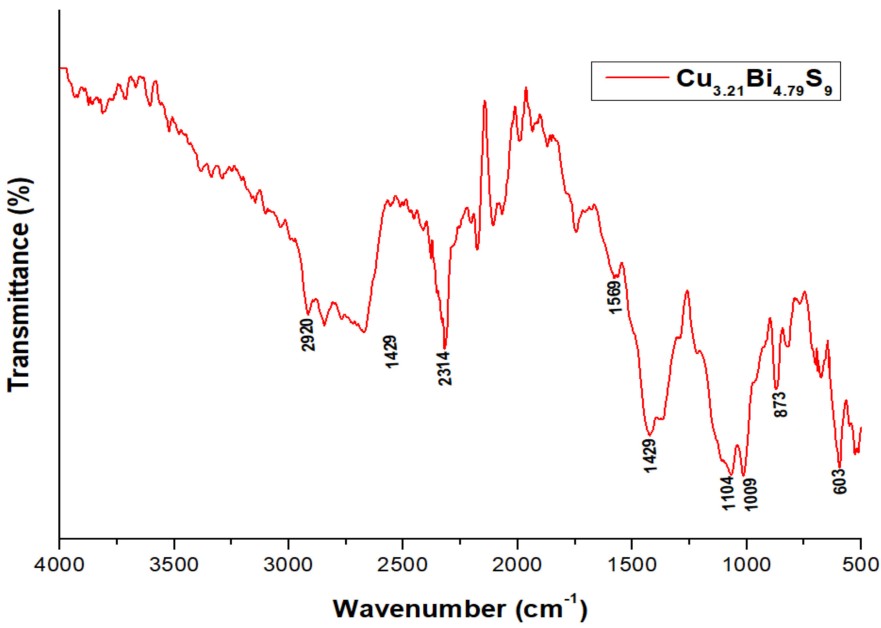

**Figure A1.** The FTIR spectrum of oleylamine-capped $Cu_{3.21}Bi_{4.79}S_9$.

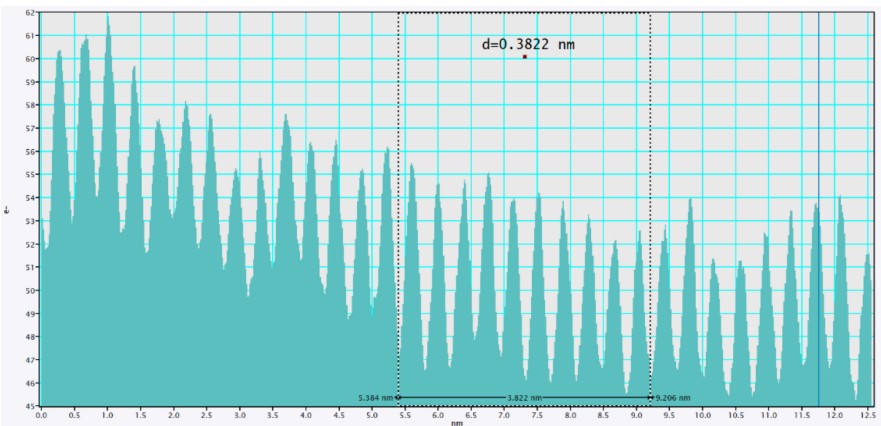

**Figure A2.** d-spacing from the HRTEM micrograph of $Cu_{3.21}Bi_{4.79}S_9$.

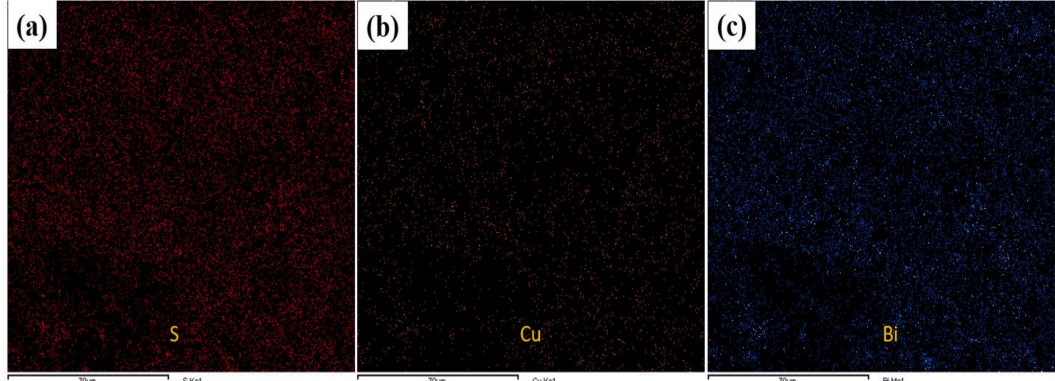

**Figure A3.** Elemental mapping of $Cu_{3.21}Bi_{4.79}S_9$. It shows the distribution of (**b**) copper, (**c**) bismuth and (**a**) sulphur in the ternary sulphide.

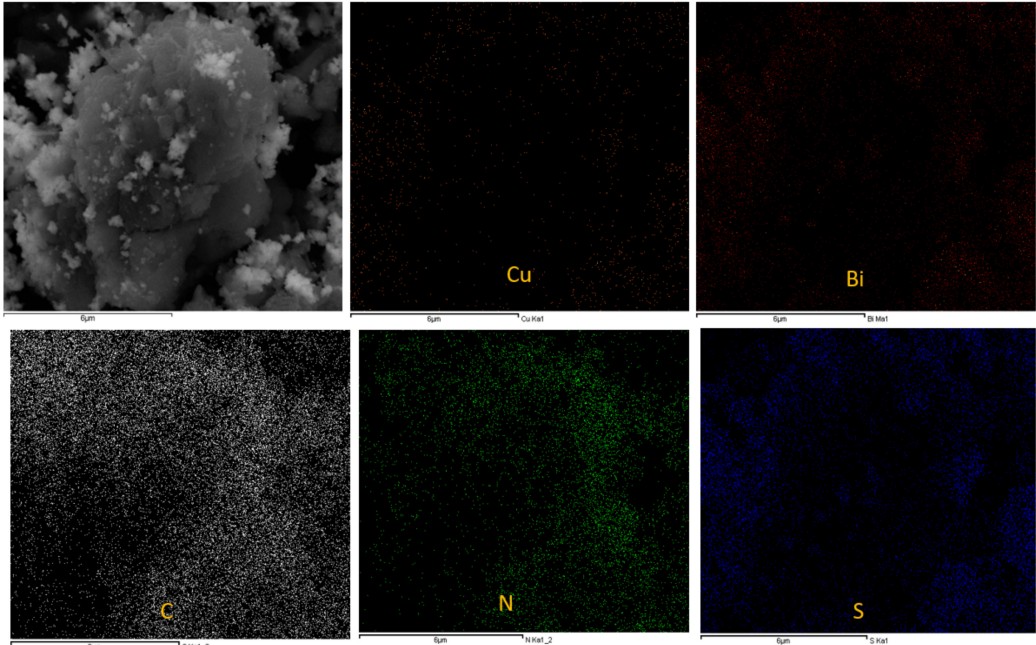

**Figure A4.** Elemental mapping of $Cu_{3.21}Bi_{4.79}S_9$/g-$C_3N_4$ nanocomposite. It shows the distribution of copper, bismuth, sulphur, carbon and nitrogen in the nanocomposite.

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
