# Peer review of "Photocatalytic Reduction of Hexavalent Chromium Using Cu3.21Bi4.79S9/g-C3N4 Nanocomposite"

_catalysts, doi:10.3390/catal12101075_

Round 1

Reviewer 1 Report

The authors reported the Cu3.21Bi4.79S9/gC3N4 nanocomposites as photocatalysts for Cr reduction. It is an interesting work . However, there are many flaws in this manuscript, including missing references and control experiments, some data is not clearly presented and even might be inaccurate, and the writing of the introduction also need some revisions. Hence, a major revision must be made. The authors must address the following questions before I can reconsider whether it can be published.

1. The figures have a lot of formatting problem. The Figure 3,4 only have half that can be seen.

2. Is that possible to give the element mapping for Figure 3? If no, the authors could give the reason and it is acceptable.

3. Figure 1, please give the XRD of both Cu3.21Bi4.79S9 and gC3N4.

4. The FTIR of Cu3.21Bi4.79S9 should be added in Figure 2.

5. Many typos. For example, Page 6, line 207, “Figure”, which Figure? Line 226, “Figure xx”. There are many others, please double check the whole manuscript.

6. For introduction, the advantages of the nanoparticle/ polymeric matrix photocatalysts need more discussion. Only narrow the polymer materials to gC3N4 cannot give the big picture to the readers, and a bit more about nanoparticle/ polymeric matrix must be discussed.  I agree with the authors this is a very important type of photocatalysts, as it would leverage the advanced optical/ electrical properties from both inorganic and organic parts. Besides, some recent important examples of this type of photocatalysts (https://doi.org/10.1021/acsmaterialslett.1c00785; https://doi.org/10.1039/D1TA07733C) must be cited and briefly discussed.

7. Page 7, I did not see “Figure S1”. Are you referring to Appendix? If so, please  put the Appendix data into supporting information.

8. What is the ratio of different elements obtained from EDX in Figure 3? A table can be made to summarize these data.

9. Figure 5a, the data is abnormal. The absorption of gC3N4 is very different from many previous reports. No absorption from 500-900 nm should be observed. And for all the curves, seems like there are a lot of scattering (for the Cu3.21Bi4.79S9, the curve is going up from 400 to 900 nm)

10. How to get the Tauc plot (equations and some description) must be added in the Methods section. Are the materials direct or indirect band gaps? Any evidence?

Author Response

Manuscript ID: Manuscript ID: catalysts-1869742

Title: Photocatalytic reduction of hexavalent chromium using Cu3.21Bi4.79S9/g-C3N4 composite

First Author: Timothy O. Ajiboye,

Corresponding Author: Damian C. Onwudiwe

REVIEWER 1

The manuscript describes the synthesis of composite photocatalysts based on g-C3N4 photocatalytic reduction of heavy metal ions and oxidation of dyes. In principle, the work is quite interesting, but the implementation is very weak, the experiments are poorly organized, there is no discussion as such, there is only a listing of various facts. In its current state, the article should be rejected.

AUTHORS RESPONSE:

The authors appreciate the reviewer for acknowledging that the work is quite interesting. To improve the implementation and the organization of the experiments, further results have been added to the introduction and the characterization section of the manuscript.

REVIEWER 1

Main comments:

All figures are of a very bad quality and should be improved.

AUTHORS RESPONSE:

The Figures have been checked and replaced with new Figures having improved quality. The anomalies that occurred when the manuscript was transferred into the mdpi template have been corrected.

REVIEWER 1

The characterization of the sample is clearly not enough, for example, there is no XPS, while the TEM is of very poor quality and does not provide information about the structure of the samples.

AUTHORS RESPONSE:

The characterization section has been improved with results such as the FTIR of the pristine copper bismuth sulphide, XRD of graphitic carbon nitride and XRD of pristine graphitic carbon nitride. These will help in understanding the formation of the nanocomposite. The importance of XPS in the characterization is to give information on the actual composition and the surface characteristics of the nanocomposite. This is the same information that has been obtained from the EDS and the zeta potential.

REVIEWER 1

The approximation was carried out with a very low accuracy and its meaning is not clear.

AUTHORS RESPONSE:

The accuracy of all the data used for drafting the manuscript have been checked again and confirmed correct.

REVIEWER 1

The authors do not explain in any way the increase in the photocatalytic activity of the composite sample.

AUTHORS RESPONSE:

The entire section 4.1 was used to explain the increase in the photocatalytic activity of the composite sample against the pristine graphitic carbon nitride and copper bismuth sulphide. The observed increase was backed up with explanations and relevant literature references.

Reviewer 2 Report

The manuscript describes the synthesis of composite photocatalysts based on g-C3N4 photocatalytic reduction of heavy metal ions and oxidation of dyes. In principle, the work is quite interesting, but the implementation is very weak, the experiments are poorly organized, there is no discussion as such, there is only a listing of various facts. In its current state, the article should be rejected.

Main comments:

1. All figures are of a very bad quality and should be improved.

2. The characterization of the sample is clearly not enough, for example, there is no XPS, while the TEM is of very poor quality and does not provide information about the structure of the samples.

3. The approximation was carried out with a very low accuracy and its meaning is not clear.

4. The authors do not explain in any way the increase in the photocatalytic activity of the composite sample.

Author Response

REVIEWER 2

The authors reported the Cu3.21Bi4.79S9/gC3N4 nanocomposites as photocatalysts for Cr reduction. It is an interesting work . However, there are many flaws in this manuscript, including missing references and control experiments, some data is not clearly presented and even might be inaccurate, and the writing of the introduction also need some revisions. Hence, a major revision must be made. The authors must address the following questions before I can reconsider whether it can be published.

AUTHORS RESPONSE:

The authors appreciate the reviewer for the recommendation and the time spent in reviewing our manuscript. The observations are valid and they will surely enhance the overall quality of the manuscript. The information to improve the manuscript has been included especially in the introduction and the characterization section of the manuscript.

REVIEWER 2

The figures have a lot of formatting problem. The Figure 3,4 only have half that can be seen.

AUTHORS RESPONSE:

The Figures have been checked and replaced with new Figures having improved quality. The anomalies that occurred when the manuscript was transferred into the mdpi template have been corrected as well.

REVIEWER 2

Is that possible to give the element mapping for Figure 3? If no, the authors could give the reason and it is acceptable.

AUTHORS RESPONSE:

The elemental mapping for the copper bismuth sulphide functionalized with graphitic carbon nitride and the pristine copper bismuth sulphide have been added to the supplementary section of the manuscript.

REVIEWER 2

Figure 1, please give the XRD of both Cu3.21Bi4.79S9 and gC3N4.

AUTHORS RESPONSE:

The XRD of both Cu3.21Bi4.79S9 and gC3N4 have been included in the manuscript.

REVIEWER 2

The FTIR of Cu3.21Bi4.79S9 should be added in Figure 2.

AUTHORS RESPONSE:

The FTIR of the pristine copper bismuth sulphide has been included in the manuscript.

REVIEWER 2

Many typos. For example, Page 6, line 207, “Figure”, which Figure? Line 226, “Figure xx”. There are many others, please double check the whole manuscript.

AUTHORS RESPONSE:

The typographical errors have been corrected in the entire manuscript.The manuscript was corrected by the language expert before resubmission.

REVIEWER 2

For introduction, the advantages of the nanoparticle/ polymeric matrix photocatalysts need more discussion. Only narrow the polymer materials to gC3N4 cannot give the big picture to the readers, and a bit more about nanoparticle/ polymeric matrix must be discussed.  I agree with the authors this is a very important type of photocatalysts, as it would leverage the advanced optical/ electrical properties from both inorganic and organic parts. Besides, some recent important examples of this type of photocatalysts (https://doi.org/10.1039/D1TA07733C) must be cited and briefly discussed.

AUTHORS RESPONSE:

The authors appreciate the reviewer for the paper suggestion. The paper has been cited and brief discussion on the carbon organic framework described by the paper has been included in the introduction part of the manuscript.

REVIEWER 2

Page 7, I did not see “Figure S1”. Are you referring to Appendix? If so, please  put the Appendix data into supporting information.

AUTHORS RESPONSE:

The Figure was renamed as Figure A1 and it has been corrected in the text. The Appendix data are in the supporting information before the references (based on the mdpi format).

REVIEWER 2

Figure 5a, the data is abnormal. The absorption of gC3N4 is very different from many previous reports. No absorption from 500-900 nm should be observed. And for all the curves, seems like there are a lot of scattering (for the Cu3.21Bi4.79S9, the curve is going up from 400 to 900 nm)

AUTHORS RESPONSE:

There was no absorption from 500-900 nm in the absorption of graphitic carbon nitride (graphitic carbon nitride is the blue line in the Figure). Similar result was obtained by Miller et al.,( https://doi.org/10.1039/C7CP02711G) and the work was properly cited in the manuscript.

REVIEWER 2

How to get the Tauc plot (equations and some description) must be added in the Methods section. Are the materials direct or indirect band gaps? Any evidence?

AUTHORS RESPONSE:

The equation and description of Tauc plot has been included in the manuscript and type of band gap identified.

Round 2

Reviewer 1 Report

I can not see the authors' response to my question somehow. But from the revised manuscript the authors has improved the quality. The current is good to publish.

Author Response

Authors appreciate the Reviewer for the kind recommendation of our manuscript for publication

Reviewer 2 Report

The authors tried to answer the comments of the reviewers, but some of the questions were not answered.

1. All figures are still of rather poor quality, authors should look at other articles in the magazine and understand how to arrange drawings.

2. Figures 8-12. Error bars should be shown for all experimental points.

3. Also, I still don't understand the need for approximation by the proposed model. "The accuracy of all the data used for drafting the manuscript have been checked again and confirmed correct." is not an answer, with R2 = 0.7114 or 0.7088, it is a very bad approximation!
